# Cost-effectiveness of the TherMax blood warmer during continuous renal replacement therapy

Michael J. Blackowicz[1]*, Max Bell[2,3], Jorge Echeverri[1], Kai Harenski[1], Marcus E. Broman[4]

1 Baxter Healthcare Corporation, Deerfield, Illinois, United States of America, 2 Department of Perioperative Medicine and Intensive Care, Karolinska University Hospital, Solna, Stockholm, Sweden, 3 Department of Physiology and Pharmacology, Karolinska Institutet, Stockholm, Sweden, 4 Perioperative and Intensive Care, Skåne University Hospital, Lund, Sweden

* michael_blackowicz@baxter.com

**Data Availability Statement:** The study data includes deidentified patients medical records. However, since CRRT treatment is uncommon, this increases the possibility to identify a certain patient

## Abstract

Hypothermia is a common adverse event during continuous renal replacement therapy (CRRT), affecting multiple organ systems and increasing risk of poor health outcomes among patients with acute kidney injury (AKI) undergoing CRRT. TheraMax blood warmers are the next generation of extracorporeal blood warmers which reduce risk of hypothermia during CRRT. The purpose of this study is to elucidate the potential health economic impacts of avoiding CRRT-induced hypothermia by using the novel TherMax blood warming device. This study compares health care costs associated with use of the new TherMax blood warmer unit integrated with the PrisMax system compared to CRRT with a standalone blood warming device to avoid hypothermia in continuous renal replacement therapy (CRRT). An economic model was developed in which relevant health states for each intervention were normothermia, hypothermia, discharge, and death. Clinical inputs and costs were obtained from a combination of retrospective chart review and publicly available summary estimates. The proportion of AKI patients treated with CRRT who became hypothermic (<36˚C) during CRRT treatment was 34.5% in the TherMax group compared to 71.9% in the 'standalone warmer' group. Given the 78.7-year average life expectancy in the US and the assumed average patient age at discharge/death of 65.4 years, the total life-years gained by avoiding mortality related to hypothermia was 9.0 in the TherMax group compared to 8.0 in the 'standalone warmer' group. Cost per life-year gained was $8,615 in the TherMax group versus $10,115 in the 'standalone warmer' group for a difference of -$1,501 favoring TherMax. The incremental cost-effectiveness ratio was negative, indicating superior cost-effectiveness for TherMax versus 'standalone warmer'. The TherMax blood warming device used with the PrisMax system is associated with lower risk of hypothermia, which our model indicates leads to lower costs, lower risk of mortality due to hypothermia, and superior cost-effectiveness.

based on their admission date, treatment duration, and discharge date. Therefore, the medicolegal department at the Skåne University Hospital has recommended to not release the data for public availability. To avoid disclosure of protected health information, all data must be maintained within Skåne University Hospital databases and the data owner is Dr. Marcus Broman, an author on this study. Dr. Mikael Bodelsson of Skåne University Hospital has agreed to serve as the contact for data accessibility requests related to this study. He can be reached at: mikael.bodelsson@med.lu.se.

**Funding:** This study was funded by Baxter Healthcare and Skåne University Hospital in the form of consultancy grants for authors Bell and Broman. Baxter Healthcare also provided support in the form of salaries for authors Blackowicz, Echeverri, and Harenski. The specific roles of these authors are articulated in the 'author contributions' section. The funders had no role in the study design, data collection and analysis, decision to publish, or preparation of the manuscript.

**Competing interests:** The authors have read the journal's policy and have the following competing interests: Authors Blackowicz, Echeverri, and Harenski are salaried employees of Baxter Healthcare with ownership interest. This does not alter our adherence to PLOS ONE policies on sharing data and materials. PrisMax, TherMax, and Prismaflex are patented products of Baxter Healthcare Corporation. Barkey Prismacomfort is a patented product of Barkey GmbH & Co. There are no other patents, products in development or marketed products associated with this research to declare.

## Introduction

The burden of acute kidney injury (AKI) during critical illness is substantial. AKI is associated with short- and long-term complications in addition to increased use of health care resources [1–6]. In the multinational AKI-EPI study over half of all patients in the intensive care unit (ICU) had AKI and 23.5% required renal replacement therapy, translating to 13.5% of all ICU patients [2].

Hypothermia is a common adverse event during continuous renal replacement therapy (CRRT), historically reported in 44% of all cases [7]. Hypothermia affects multiple organ systems. It increases risks of bradycardia and arrhythmia, is associated with coagulopathy and ensuing transfusion requirements, and can impair pharmacodynamics [8]. In a large meta-analysis of 10,834 patients from 42 studies, sepsis mortality was significantly higher in patients with hypothermia than those with normothermia and fever [9]. A study from 2020 by Shimazui and co-workers showed hypothermia to be coupled with increased mortality in non-elderly septic patients [10]. Interestingly, in responding to questions regarding whether CRRT was a confounder in this study [11], the authors performed additional analyses and concluded that hypothermia was still associated with mortality, including in patients treated with CRRT. Hypothermia has been associated with significantly longer ICU length of stay in trauma patients treated in the ICU [12]. A study describing the effect of cooling on critically ill febrile patients indicated that hypothermia induced by CRRT results in immune system dysfunction [13] and suggested that assessment of long-term outcomes in this population is needed.

In a recent study, Bell and co-workers found that hypothermia was less prevalent in patients treated with the latest generation CRRT platform, the PrisMax system and the integrated TherMax blood warmer, compared to a previous CRRT platform with a standalone blood warmer [14]. The warmer has its control software integrated into the PrisMax system that has operator-selectable temperature settings. The TherMax device directly warms the blood by heating plates, adjusting warming based on estimated return blood temperature to achieve prescribed temperature targets. A bi-directional connection with the PrisMax system allows the warmer to respond to changing treatment parameters. In contrast, the current standard of care typically involves various temperature control methods including clinician-prescribed warming of the blood or fluids using standalone blood/fluid warmers that work independently from the treatment parameters of the CRRT device.

Cost-effectiveness analyses for health technologies are becoming increasingly important with increasing and often more costly treatment options over time. They provide information on where cost-burdens can be reduced to transfer resources to and increase the understanding of health care efficacy. The present investigation aims to elucidate the potential health economic impacts of avoiding CRRT-induced hypothermia. Specifically, the objectives are to: (1) estimate and compare the risk of hypothermia during the use of the TherMax blood warmer on the latest generation CRRT platform versus a previous generation CRRT blood warmer and (2) estimate and compare costs and cost-effectiveness of the TherMax warmer versus a previous generation blood warmer.

## Materials and methods

### Patient population

The Adult Intensive Care Department at Skåne University Hospital, Lund, Sweden is a tertiary mixed surgical and medical unit with about 1000 admissions per year and nine beds. All provider specialties are present 24/7. Prior to November 2018, patients requiring dialysis for AKI were treated with CRRT via the PrismaFlex device with Barkey warmer (Baxter Healthcare).

After November 2018, PrismaFlex devices were phased out and replaced with PrisMax system with integrated TherMax blood warmer unit (Baxter Healthcare). Patients were included in the study if they were: (1) admitted to the Skåne University Hospital's Adult ICU between December 2006 and September 2020; (2) treated with CRRT for AKI for at least 1 hour; and (3) had normal body temperature (36.5˚C-37.5˚C) at the start of CRRT. Patients were excluded if they had undergone induced/therapeutic hypothermia, were treated with more than one CRRT device in a single ICU stay, or if the CRRT device used was not ascertainable.

Patient body temperature and CRRT device settings were stored in the Intellispace Critical Care and Anesthesia (ICCA) system. The temperature value set by the warmer and the patient body temperature were recorded hourly and stored automatically in the ICCA system, along with any relevant notes regarding patient status and therapy. Temperatures were recorded from an esophageal probe, the indwelling catheter in the urinary bladder in a majority of cases, and further from the thermistor of the pulse contour cardiac output monitoring system (Pulsion Medical Systems, Munich, Germany), or a pulmonary artery catheter (Edwards Lifesciences, Irvine, CA). For a very few minority of stable patients, tympanic temperature recordings were taken manually at the bedside. These methods are all accurate within 0.1 Celcius. This study was approved by the Regional Ethical Board of Southern Sweden (Dnr 2020–04642). Informed consent was not required since data did not contain uniquely identifying information and were analyzed anonymously.

## Health outcomes and costs

Hypothermia risk was calculated by warmer group from patient-level data as the proportion of patients who were hypothermic during CRRT treatment. Hypothermia was defined as body temperature (measured as described previously) < 36˚C for at one or more hourly readings during CRRT treatment within a single ICU stay.

Mortality risk among normothermic AKI patients treated with CRRT in the ICU was ascertained from a literature-based estimate of mortality among AKI patients treated with CRRT in the ICU [15]. Mortality risk among hypothermic patients was ascertained by multiplying the estimate obtained above by a literature-based estimate of SAPS-2 adjusted relative risk of ICU mortality in hypothermic patients versus normothermic patients [16]. The latter study did not exclusively include AKI patients treated with CRRT, but was limited to patients who had normal body temperature (36.5˚C-37.5˚C) at admission, excluding patients who reached a state of hyperthermia (>37.5˚C) and patients who had undergone intentionally induced/therapeutic hypothermia. Average life-years gained per patient was calculated as the difference in average life years remaining weighted by sex and by proportion of patients on long-term dialysis [17] and the average patient age at discharge/death (Table 1) [18,19].

The duration of CRRT was assumed to be 7 days for both warmer types since the warmer is unlikely to impact the duration of dialysis [20]. Average ICU length of stay (LOS) among normothermic ICU patients was ascertained from a literature-based estimate among patients with AKI treated with CRRT in an international cohort of 1006 CRRT patients [20]. Average ICU LOS among hypothermic patients was ascertained by adding a literature-based estimate of mean difference in ICU LOS between hypothermic and normothermic trauma patients requiring ICU admission [12].

The average cost per patient per day of CRRT activity and the average cost per ICU day were ascertained from a previously published cost-effectiveness analysis of CRRT versus IHD in the US [15]. The average cost per ICU day was ascertained from a previously published study of ICU costs in the US [21]. The model also includes device costs for PrisMax with integrated TherMax warmer as well as PrismaFlex with standalone Barkey warmer as a

**Table 1. Baseline demographic and clinical characteristics.**

| Characteristic | TherMax Warmer | Standalone Warmer | p-value |
|---|---|---|---|
| Number of treatments | 29 | 526 | |
| Age (years)[a] | 65.39 ± 12.53 | 65.45 ± 14.66 | 0.399 |
| Sex (female) | 42.5% | 40.9% | 0.393 |
| SAPS3[a] | 74.09 ± 13.63 | 75.72 ± 14.85 | 0.325 |
| KDIGO | 3 | 3 | 1.000 |
| Blood flow rate (ml/min)[a] | 184.45 ± 50.40 | 136.64 ± 33.89 | <0.001 |
| Effluent flow rate (ml/min)[a] | 42.18 ± 26.72 | 63.44 ± 83.99 | 0.003 |
| Hemoglobin (g/L)[a] | 108.48 ± 22.17 | 110.62 ± 19.76 | 0.348 |

[a] Mean ± SD.

comparison. Device costs were translated from published US list prices [22] to per-patient device cost assuming an average of 42 patients per year based on seven CRRT days per patient over a 6-year lifespan with 80% use/efficiency.

## Health states and model form

All comparisons were made between the latest generation PrisMax device with the integrated TherMax warmer (Nov 2018 to Dec 2020) versus previous generation warmers, represented by PrismaFlex with a standalone Barkey warmer (Jan 2006 to Nov 2018). For each CRRT warmer type, the relevant health states were normothermia, hypothermia, discharge, and death. Clinical inputs and costs are listed in Tables 2 and S1. It was assumed there is no additional cost associated with hypothermia other than the costs related to increased length of stay in the ICU.

**Table 2. Primary model inputs.**

| Parameter | TherMax Warmer | Standalone Warmer |
|---|---|---|
| **Patient Age (yr)** | 65.4 | 65.4 |
| **Life years remaining (US population age 65)** | 17.5 | 17.5 |
| **Life years remaining (ESRD on dialysis age 65)** | 7.8 | 7.8 |
| **Chronic Dialysis (%)** | 21.8 | 21.8 |
| **Hypothermia (%)** | 34.5 | 71.9 |
| **Mortality (%)** | | |
| Among normothermic | 40.0 | 40.0 |
| Among hypothermic | 59.2 | 59.2 |
| **ICU Length of Stay (days)** | | |
| Among normothermic | 12.0 | 12.0 |
| Among hypothermic | 13.7 | 13.7 |
| **Duration of CRRT (days)** | 7.0 | 7.0 |
| **Costs** | | |
| Device Cost (US list price) | $50,000[a] | $38,000[b] |
| Cost per inpatient day | $5,611 | $5,611 |
| Cost of CRRT per day | $1,047 | $1,047 |

[a] US list price for PrisMax [22].
[b] US list price for PrismaFlex [22].

Due to uncertainty around the inputs, a deterministic sensitivity analysis was performed to assess the impact of each estimate on the results. The sensitivity analysis assessed the impact of setting ICU LOS equal between hypothermic and normothermic patients, as well as ±1 day for each group. All other inputs were allowed to vary by ±20%. Results were displayed graphically in a tornado plot.

## Results

After November 2018, there were 29 ICU admissions of patients treated with CRRT using the PrisMax system with the TherMax blood warmer for a median of 68 hours (9.7 days) per admission. Prior to November 2018, 526 patients were treated with CRRT using the Prismaflex system with the previous generation of standalone blood warmer for a median of 62 hours (8.9 days) per admission. Characteristics for the TherMax warmer group and the standalone warmer group are presented in Table 1.

The proportion of AKI patients treated with CRRT who became hypothermic at any point during CRRT treatment was 34.5% in the TherMax group compared to 71.9% in the 'stand-alone warmer' group. Given the 78.7-year average life expectancy in the US and the assumed average patient age at discharge/death of 65.4 years, the total life-years gained by avoiding mortality related to hypothermia was 9.0 in the TherMax group compared to 8.0 in the 'stand-alone warmer' group (Table 3).

The total cost of ICU stay was $70,664 in the TherMax group versus $74,274 in the 'stand-alone warmer' group for a difference of $3,562 favoring TherMax, while cost of CRRT activity was assumed to be equal at $7,344/admission Device cost per patient was $200 for TherMax powered by Prismax versus $152 for the 'standalone warmer', powered by PrismaFlex. Cost per life-year gained was $8,615 in the TherMax group versus $10,115 in the 'standalone warmer' group for a difference of -$1,501 favoring TherMax. The incremental cost-effectiveness ratio was -$3,619, indicating superior cost-effectiveness for TherMax versus the 'stand-alone warmers' (Table 3).

The inputs with the largest impact on cost difference between TherMax and the standalone warmers were ICU LOS for patients with hypothermia followed by ICU LOS for patients with normothermia. Increasing the ICU LOS for hypothermia or decreasing for normothermia resulted in a higher cost due to a larger difference in ICU LOS between patients with and without hypothermia. Setting the ICU LOS equal between patients with and without hypothermia (either 12 days or 13.7 days) resulted in no cost difference overall (Fig 1). Differential risk of CRRT-induced hypothermia between the TherMax warmer and the standalone warmers also played a role in the model as did cost of an ICU day, but even after allowing for 20% variability on each input, the TherMax warmer was still a cost-saving technology overall.

**Table 3. Cost-effectiveness results.**

| Summary Statistic | TherMax Warmer | Standalone Warmers | Difference |
|---|---|---|---|
| **Total Costs** | $77,590 | $81,152 | -$3,562 |
| Cost of ICU Stay | $70,664 | $74,274 | -$3,610 |
| Cost of CRRT | $6,727 | $6,727 | $0 |
| Device Cost[a] | $200 | $152 | $48 |
| **Life Years Gained** | 9.0 | 8.0 | 1 |
| **Total Costs per LYG** | $8,615 | $10,115 | -$1,501 |
| **ICER**[b] | -$3,619 | - | - |

[a] Device cost calculated as: (device list price)/(365 days per year)/(7 days per patient) * (80% use efficiency).

[b] Calculated as the difference in cost divided by the difference in LYG; Negative ICER indicates superior cost-effectiveness.

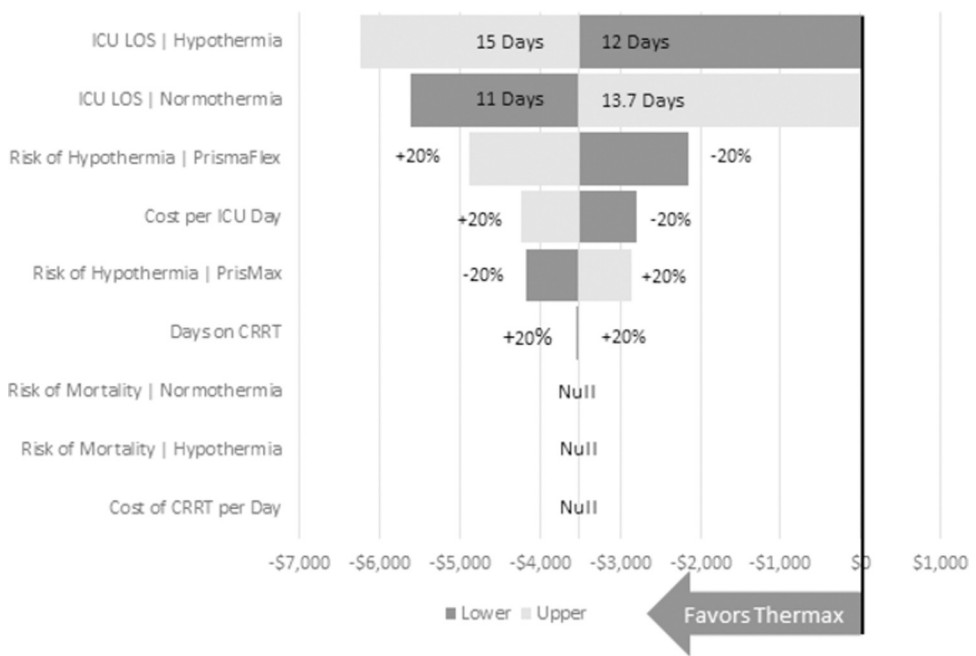

**Fig 1. Deterministic sensitivity analysis.**

## Discussion

To the best of our knowledge, this is the first economic analysis of the impact of a CRRT blood warmer. Our analysis shows that use of the new TherMax blood warmer unit integrated with the PrisMax system reduced costs of treatment in the ICU by more than $3500 compared to CRRT with a standalone warmer. The cost per life-year gained was reduced by nearly $2000. Accordingly, the incremental cost-effectiveness ratio indicates superior cost-effectiveness for the TherMax device.

Several previous studies have analyzed the economic impact of CRRT and intermittent renal replacement therapy (IRRT). Klarenbach and co-workers [23] and De Smedt et al. [24] concluded that CRRT did not provide either health or economic advantage over IRRT. In contrast to those two reports, a more recent investigation showed CRRT to have higher ICU costs but cumulatively lower total costs as compared to IRRT due to a lower occurrence of dialysis dependence among survivors [15]. That study used a Markov model with two health states for AKI survivors, dialysis dependence and dialysis independence. In a similar methodological setup from a 2019 study, with a model-based cost-utility analysis estimating long-term costs and health outcomes for a hypothetical cohort with a Markov model, CRRT dominated IRRT by cumulating more quality-adjusted life years and less costs [25].

While published literature is absent of specific studies on the economic impact of hypothermia, there are multiple reports of the association of low body temperature with detrimental outcomes. In 2020, Park et al. presented data on emergency department patients with sepsis; [26] in-hospital mortality rates in the hyperthermia, normothermia, and hypothermia groups were 8.5%, 20.6%, and 30.8%, respectively (p < 0.001). In a multivariate analysis, compared with hyperthermia, even normothermia was significantly associated with an increased in-hospital mortality. Data from Japan show that in patients with severe sepsis, the proportions of ICU-free and ventilator-free days were significantly smaller in patients with hypothermia. Further, hypothermia has been shown to be associated with a significantly higher disease severity, mortality risk, and lower implementation of sepsis bundles [27].

The relationship between body temperature and outcomes among critically ill patients is exceedingly well documented. Hypothermia may be related to high mortality in patients with sepsis [27–33]. However, it is unknown if *inadvertent* hypothermia, *induced* by the CRRT circuit carries similar risks to those described above. Specifically, we do not have data that fully elucidate the potential association between induced hypothermia and mortality risk. For instance, using studies of *targeted* hypothermia after cardiac arrest [34] is not feasible for obvious reasons. In the absence of focused studies on inadvertent hypothermia, we used data from Ethgen [15] and Erkens [16].

This study has both strengths and limitations. First, this is the first study attempting to calculate the economic impact of CRRT associated hypothermia. The underlying data, previously reported [14], consists of over 55,000 measured temperatures from over 300 individuals. It included information on both historical and current CRRT systems and their respective warming devices. Data granularity and validity is very high and have been described before [14,35], specifically with automatic data transfer without manual input by healthcare staff. Moreover, with regards to the economic assessments, we have consistently used conservative assumptions, and have added sensitivity analyses in order to adequately gauge the impact of our calculations.

The limitations of the study include: 1) the model relies on data regarding CRRT duration, ICU LOS, and health outcomes from studies that did not specifically evaluate inadvertent CRRT-induced hypothermia; 2) the dataset included a large set of measured temperatures, but the historic group was larger and spanned over 12 year while the current PrisMax/TherMax data was obtained over 10 months; 3) multiple forms of health care economic systems exist in the industrialized countries, making precise adaptations, or "translations", to singular countries difficult; and 4) classification and inclusion of patients with short duration (<1 hour) of low-risk/short-duration patients into a binary (yes/no) CRRT exposure may introduce misclassification, but misclassification is expected to be non-differential with respect to warmer type or time period and unlikely to introduce bias into the results except possibly toward null association.

## Conclusions

In conclusion, switching to the TherMax blood warmer with the PrisMax system for CRRT is associated with less risk of hypothermia. Given the risk of mortality associated with hypothermia, our model also indicates that the switch reduced the risk of mortality associated with hypothermia, which lead to more life-years and superior cost-effectiveness of the PrisMax with integrated TherMax blood warmer for CRRT compared to the previous generation of devices with standalone warmers.

## Supporting information

**S1 Table. Economic model inputs and sources.** This supporting table is an more detailed expansion of the model inputs list provided in Table 1 of the main text. This table additionally includes sources for each input along with a description of the source and statistic. (DOCX)

## Author Contributions

**Conceptualization:** Michael J. Blackowicz, Max Bell, Jorge Echeverri, Kai Harenski, Marcus E. Broman.

**Data curation:** Michael J. Blackowicz, Marcus E. Broman.

**Formal analysis:** Michael J. Blackowicz, Marcus E. Broman.

**Funding acquisition:** Jorge Echeverri, Kai Harenski.

**Investigation:** Michael J. Blackowicz, Max Bell, Marcus E. Broman.

**Methodology:** Michael J. Blackowicz, Max Bell.

**Project administration:** Michael J. Blackowicz, Jorge Echeverri, Kai Harenski.

**Resources:** Max Bell, Jorge Echeverri, Kai Harenski, Marcus E. Broman.

**Software:** Michael J. Blackowicz.

**Supervision:** Jorge Echeverri, Kai Harenski.

**Validation:** Michael J. Blackowicz, Marcus E. Broman.

**Visualization:** Michael J. Blackowicz.

**Writing – original draft:** Michael J. Blackowicz, Max Bell, Marcus E. Broman.

**Writing – review & editing:** Michael J. Blackowicz, Max Bell, Jorge Echeverri, Kai Harenski, Marcus E. Broman.

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
