## [Decision Letter · Decision Letter 0]

2 Jun 2021

PONE-D-21-12993

Cost-effectiveness of the TherMax Blood Warmer During Continuous Renal Replacement Therapy

PLOS ONE

Dear Dr. Blackowicz,

Thank you for submitting your manuscript to PLOS ONE. After careful consideration, we feel that it has merit but does not fully meet PLOS ONE’s publication criteria as it currently stands. Therefore, we invite you to submit a revised version of the manuscript that addresses the points raised during the review process.

We look forward to receiving your revised manuscript.

Kind regards,

Andrea Ballotta

Academic Editor

PLOS ONE

Journal Requirements:

3. Thank you for providing the following Funding Statement: 

'The study was funded by consultancy grants for authors Bell and Broman from Baxter Healthcare and from Skåne University Hospital. Authors Blackowicz, Echeverri, and Harenski are full time employees of Baxter Healthcare with ownership interest. All authors played a role in study design and manuscript preparation. Data was collected independently of the Baxter Healthcare.'

a. We note that one or more of the authors is affiliated with the funding organization, indicating the funder may have had some role in the design, data collection, analysis or preparation of your manuscript for publication; in other words, the funder played an indirect role through the participation of the co-authors.

If the funding organization did not play a role in the study design, data collection and analysis, decision to publish, or preparation of the manuscript and only provided financial support in the form of authors' salaries and/or research materials, please review your statements relating to the author contributions, and ensure you have specifically and accurately indicated the role(s) that these authors had in your study in the Author Contributions section of the online submission form. Please make any necessary amendments directly within this section of the online submission form.  Please also update your Funding Statement to include the following statement: “The funder provided support in the form of salaries for authors [insert relevant initials], but did not have any additional role in the study design, data collection and analysis, decision to publish, or preparation of the manuscript. The specific roles of these authors are articulated in the ‘author contributions’ section.”

If the funding organization did have an additional role, please state and explain that role within your Funding Statement.

Within your Competing Interests Statement, please confirm that this commercial affiliation does not alter your adherence to all PLOS ONE policies on sharing data and materials by including the following statement: "This does not alter our adherence to  PLOS ONE policies on sharing data and materials.” (as detailed online in our guide for authors http://journals.plos.org/plosone/s/competing-interests). If this adherence statement is not accurate and  there are restrictions on sharing of data and/or materials, please state these.

Please note that we cannot proceed with consideration of your article until this information has been declared.

5. Please include captions for your Supporting Information files at the end of your manuscript, and update any in-text citations to match accordingly. Please see our Supporting Information guidelines for more information: http://journals.plos.org/plosone/s/supporting-information

Additional Editor Comments:

Tx for your contribution. As stated by the reviewer the paper needs major revision

Reviewers' comments:

Reviewer's Responses to Questions

**Comments to the Author**

1. Is the manuscript technically sound, and do the data support the conclusions?

Reviewer #1: Partly

2. Has the statistical analysis been performed appropriately and rigorously? 

Reviewer #1: Yes

3. Have the authors made all data underlying the findings in their manuscript fully available?

Reviewer #1: Yes

4. Is the manuscript presented in an intelligible fashion and written in standard English?

Reviewer #1: Yes

5. Review Comments to the Author

Reviewer #1: The authors described a novel device to reduce risk of hypothermia during CRRT, the topic is interesting in the ICU setting, perhaps the manuscript needs major revision and the conclusions are not completely supported by data.

Major revision:

- Please move the definition of hypothermia in the methods.

- Please describe in the methods how data transfer has been made.

- Is there any chance to record the temperature every minute? Did the authors consider in the analysis a single value recorded or at least an hour of hypothermia?

- Page 4 lines 90-91: The authors mentioned as inclusion criteria CRRT for at least 1 hour, I think the duration of the treatment is too short to affect the outcome, please discuss about that.

-Please provide the mean duration of the CRRT in the 2 groups.

- Page 5, line 98: Recording temperature from different sites could be a limit of the study, please discuss about that.

- Table 1: Please write the exact p value instead of “not significant” or “p<0.05”.

- Table 1: Number of treatments in the 2 groups are not in accordance with the text, in the table are 310 vs 32, in the manuscript the authors mentioned 526 patients treated with CRRT, it’s difficult for me to understand.

- Table 1: The comparison between 310 and 32 treatments could be a statistical issue.

- In the introduction the authors described the potential complications of the hypothermia, such as arrhythmia, coagulopathy, but they did not mention kind and rate of complications in the study.

- Page 11: It’s difficult for me to understand this limit “the current data was obtained over 10 months”, the authors wrote in the methods that they included patients between 2006 and 2020, please clarify.

- In the conclusion: the statement “TherMax reduced the risk of mortality associated with hypothermia” is not supported by data, in table 2 I can see the same mortality for hypothermia in both groups 59.2% and I think it’s quite hard to assume that the mortality is due to hypothermia, please make a comment.

Minor revision:

- Please type “TherMax” in the same way in the manuscript (“Thermax” vs “TherMax”).

6. PLOS authors have the option to publish the peer review history of their article (what does this mean?). If published, this will include your full peer review and any attached files.

Reviewer #1: **Yes: **Camilla L'Acqua

---

## [Author Response · Author response to Decision Letter 0]

28 Sep 2021

Thank you for your insight. We have addressed all concerns and included them in an separately uploaded document, as requested by the editor.

---

## [Decision Letter · Decision Letter 1]

12 Jan 2022

Cost-effectiveness of the TherMax blood warmer during continuous renal replacement therapy

PONE-D-21-12993R1

Dear Dr. Blackowicz,

We’re pleased to inform you that your manuscript has been judged scientifically suitable for publication and will be formally accepted for publication once it meets all outstanding technical requirements.

Kind regards,

Andrea Ballotta

Academic Editor

PLOS ONE

Additional Editor Comments (optional):

Thank you again for your contribution and i apologize for the delay. On the basis of the reviewer's comments i deem the paper suitable for publication.

Reviewers' comments:

Reviewer's Responses to Questions

**Comments to the Author**

1. If the authors have adequately addressed your comments raised in a previous round of review and you feel that this manuscript is now acceptable for publication, you may indicate that here to bypass the “Comments to the Author” section, enter your conflict of interest statement in the “Confidential to Editor” section, and submit your "Accept" recommendation.

Reviewer #1: All comments have been addressed

2. Is the manuscript technically sound, and do the data support the conclusions?

Reviewer #1: Yes

3. Has the statistical analysis been performed appropriately and rigorously? 

Reviewer #1: Yes

4. Have the authors made all data underlying the findings in their manuscript fully available?

Reviewer #1: Yes

5. Is the manuscript presented in an intelligible fashion and written in standard English?

Reviewer #1: Yes

6. Review Comments to the Author

Reviewer #1: (No Response)

7. PLOS authors have the option to publish the peer review history of their article (what does this mean?). If published, this will include your full peer review and any attached files.

Reviewer #1: No

---

## [Editor Report · Acceptance letter]

21 Jan 2022

PONE-D-21-12993R1 

Cost-effectiveness of the TherMax blood warmer during continuous renal replacement therapy 

Dear Dr. Blackowicz:

I'm pleased to inform you that your manuscript has been deemed suitable for publication in PLOS ONE. Congratulations! Your manuscript is now with our production department. 

Kind regards, 

on behalf of

Dr. Andrea Ballotta 

Academic Editor

PLOS ONE